

# Genetic diversity of the Hungarian Gidran horse in two mitochondrial DNA markers

Nikolett Sziszkosz, Sándor Mihók, András Jávor and Szilvia Kusza

Institute of Animal Science, Biotechnology and Nature Conservation, University of Debrecen, Debrecen, Hungary

## ABSTRACT

The Gidran is a native Hungarian horse breed that has approached extinction several times. Phylogenetic analysis of two mitochondrial markers (D-loop and cytochrome-b) was performed to determine the genetic characterization of the Gidran for the first time as well as to detect errors in the management of the Gidran stud book. Sequencing of 686 bp of *CYTB* and 202 bp of the D-loop in 260 mares revealed 24 and 32 haplotypes, respectively, among 31 mare families. BLAST analysis revealed six novel *CYTB* and four D-loop haplotypes that have not been previously reported. The Gidran mares showed high haplotype (*CYTB*: $0.8735 \pm 0.011$; D-loop: $0.9136 \pm 0.008$) and moderate nucleotide (*CYTB*: $0.00472 \pm 0.00017$; D-loop: $0.02091 \pm 0.00068$) diversity. Of the 31 Gidran mare families, only 15 *CYTB* (48.4%) and 17 D-loop (54.8%) distinct haplotypes were formed using the two markers separately. Merged markers created 24 (77.4%) mare families, which were in agreement with the mare families in the stud book. Our key finding was that the Gidran breed still possesses high genetic diversity despite its history. The obtained haplotypes are mostly consistent with known mare families, particularly when the two mtDNA markers were merged. Our results could facilitate conservation efforts for preserving the genetic diversity of the Gidran.

Corresponding author
Szilvia Kusza, kusza@agr.unideb.hu, kuszasz@hotmail.com

## INTRODUCTION

Conservation of domestic animal breeds has played an ever increasing part in biodiversity preservation. Hungary realized relatively early the unique value of maintaining the genetic diversity of endangered species (*Bodó, 1985*). The Gidran is one of the smallest horse populations among Hungarian horses (*Pataki, 1996*). It was crossed with and influenced many other breeds in Eastern Europe and is closely associated with the Kisbér Half Breed, which is another traditional Hungarian breed. Chestnut is the usual colour of the Gidran, but other colours common to the Arab horse occur in this breed. In addition to having cultural and genetic value, the Gidran is well known for its athletic speed, agility, endurance, well-balanced temperament, and robust build (*Glyn, 1971*). Due to its unique characteristics, the Gidran is widely used in many equestrian sports, such as horse jumping and carriage driving, and achieves outstanding results in international competitions. Its origin goes back to 1816, but the Gidran was only registered as a separate breed in 1885 by the Austrian Ministry of Defence (*Jónás et al., 2006*).

During its uncertain history, two major bottleneck effects drove the breed to the edge of extinction. Fortunately, the reconstruction and regeneration of the Gidran began because of a few dedicated breeders in the 1990's. Retrenching the number of mare families was a key moment in this preservation, which took into account that each mare family has more than twenty generations of breeding history (*Jónás et al., 2006*). As a consequence, the final number of mare families was determined, and the latest version of the official Gidran stud book was published in 2005 (*Mihók, 2005*). The date of the establishment of each mare families and the name of the founder mares are summarized in the Table S5. Although, the ongoing restoration of the Gidran is a notable example of gene pool protection (*Mihók & Bodó, 2003*), the status of this horse is still endangered. According to the Domestic Animal Diversity Information System database of the Food and Agriculture Organization (FAO DAD-IS), the Hungarian Gidran population is composed of 298 mares and 21 stallions, but smaller populations also exist in Romania and Bulgaria (*Food and Agriculture Organization of the United Nations, 2014*). Therefore, maintaining Gidran biodiversity is an important challenge not only for gene preservation but also from the point of view of the World Heritage (*Mihók & Bodó, 2003*).

Within the framework of breed conservation, genetic characterization acts an important aspect of maintaining breed integrity and managing genetic resources (*Glowatzki-Mullis et al., 2006*). Mitochondrial DNA (mtDNA) analysis has been used in phylogenetic and domestication studies and displays a high level of genetic variation among maternal lineages in horses (*Achilli et al., 2012*; *Cieslak et al., 2010*; *Jansen et al., 2002*). Contrary to genomic DNA, mtDNA has several unique features: it consists of a single circular molecule, it follows almost exclusively maternal inheritance, and its replication occurs independently of the cell cycle. The horse mitochondrial genome is composed of approximately 16,660 nucleotides, which encode 13 polypeptides that form part of 4 protein complexes (CI, CIII, CIV and CV) of the OXPHOS system, 22 transfer RNAs, and 2 ribosomal RNA. Due to its strict maternal inheritance, individuals within a maternal family line should share the same mtDNA haplotypes, thereby allowing an evaluation of maternal line assignment accuracy (*Wan et al., 2004*). Several investigations have shown that using two or more mtDNA markers might be more robust and powerful for genetic diversity analysis (*Pedrosa et al., 2005*). Therefore, analysis of sequence variations of mtDNA such as *CYTB* or D-loop region is an outstanding approach for the mapping of the Gidran's maternal lineage.

The main goal of this study is to examine the genetic diversity and relations among the Gidran maternal lines. We present the first phylogenetic characterization of the Gidran for the identification of rare or distinct mtDNA haplotypes using 686 bp and 202 bp sequences of the mitochondrial *CYTB* and D-loop, respectively. The second aim of the recent study was to recognize the overlapping haplotypes or errors in the management of the stud book to gain a better understanding of the genetic variability among the Gidran mare families. Our results could complement the recent conservation strategies to maintain the genetic diversity of this traditional horse breed.

## MATERIALS AND METHODS

### Hair sample collection and DNA extractions

Hair samples were collected from 260 mares representing the 31 Gidran mare families (borodi 1, 2, 3, 5, 6, 7, 14, 17, 18, 19, mezőhegyesi 1, 2, 3, 4, 5, 6, 7, 8, 9, 11, 12, 13, 14, 15, 17, 18, 19, 21 and népies 9, 22, 23) in Hungary. The mare families of two horses (247G and 202G) were unknown. Total genomic DNA was extracted from the hair follicles according to the FAO protocol (*FAO/IAEA, 2004*). All horses in this study were client-owned on which no harmful invasive procedure was performed; and there was no animal experimentation according to the legal definitions in Europe (Subject 5f of Article 1, Chapter I of the Directive 2010/63/UE of the European Parliament and of the Council).

### PCR amplification and sequencing

Based on the reference *Equus caballus* mtDNA sequences (GenBank accession nr.: X79547 and JN398377), a 1092 bp length fragment of the cytochrome b (*CYTB*) gene was amplified using our own primer pairs designed with Primer3, which is free software available online (*Untergasser et al., 2012*). The synthesized primers were as follows: *14115F* (5′-TTCCCACGTGGAATCTAA CC-3′) and *15206R* (5′-ACTAACATGAATCGGCGGAC-3′). The 297 bp segment of the D-loop region was amplified with previously published primer pairs (*Priskin et al., 2010*). PCR amplifications were performed in 20 μL reaction volumes comprising 0.1 mM dNTPs (Thermo Fisher Scientific, Waltham, MA, USA), 5×Colorless GoTaq Flexi Buffer (Promega, Madison, WI, USA), 0.75 mM MgCl$_2$ (Promega, Madison, WI, USA) 0.125 mM of each primer (Sigma-Aldrich, St. Louis, MO, USA), 0.75 U GoTaq DNA polymerase (Promega, Madison, WI, USA), and 2 μL (40–200 ng/μL) horse DNA extract. Amplifications were carried out with MJ Research PTC-200 Thermal Cycler (MJ Research, Watertown, MA, USA). The protocol included pre-denaturation at 95 °C (10 min), followed by 35 cycles of 95 °C (30 s), 62 °C (45 s), 72 °C (30 s), and then by a final extension at 72 °C for 10 min and a 4 °C hold. For verification the length of the fragments, all PCR products (5 μL) were examined by standard agarose gel electrophoresis using 2% agarose gel and stained with Gelred (Biotium, Hayward, CA, USA. The PCR amplification were sufficient for 250 (*CYTB*), 246 (D-loop) individuals. Finally, the PCR amplicons were purified using a DNA/RNA Extraction (PCR-M Clean up System) Kit (Viogene-BioTek, Taipei, Taiwan). PCR products were commercially sequenced by Macrogen Sequencing Service (Macrogen, Amsterdam, the Netherlands), PCR primers were also used as sequencing primers. The obtained DNA sequences were compared with the reference sequences from GenBank using Clustal X (*Thompson et al., 1997*).

### Data analyses

Phylogenetic analysis was conducted for a total of 250 (*CYTB*), 246 (D-loop) individuals. Combined *CYTB* and D-loop analysis was limited to those 242 horses where the PCR amplifications were successful for both markers. BioEdit v7.2.5 (*Hall, 2004*) sequence alignment editor software was used to proof and correct individual electropherograms of the sequences. All sequence alignments were performed using a general reference sequence (GenBank accession nr.: X79547) and a latterly used reference sequence (GenBank

**Table 1** Summary of the diversity parameters and haplotypes found in the Gidran mares according to the mtDNA *CYTB* and D-loop sequences and their combination.

| mtDNA marker | Number of nucleotides | Number of animals | Number of detected haplotypes | Number of new haplotypes | Polymorphic sites | Haplotype diversity ± SD | Nucleotide diversity ± SD |
|---|---|---|---|---|---|---|---|
| CYTB | 686 | 250 | 24 | 6 | 23 | 0.8735 ± 0.011 | 0.00472 ± 0.00017 |
| D-loop | 202 | 246 | 32 | 4 | 26 | 0.9136 ± 0.008 | 0.02091 ± 0.00068 |
| *CYTB &* D-loop | 893 | 242 | 49 | – | 49 | 0.9402 ± 0.006 | 0.00837 ± 0.00020 |

**Notes.**
SD, Standard Deviation.

accession nr.: JN398377) is also used in DomeTree, wich is toolkit for mtDNA analyses in domesticated animals (*Peng et al., 2015*). Complementary sequences were assembled and truncated to a length of 686 bp (*CYTB*) and 202 bp (D-loop) to allow for maximum sample size.

A BLAST search in the NCBI database was used to determine any previously unreported haplotypes. Standard diversity measures, such as polymorphic sites (Ps), haplotype (h) and nucleotide diversity (p), were calculated by DNASP 5.0 software (*Rozas et al., 2003*). A pairwise distance matrix between the mtDNA haplotypes was independently calculated for the *CYTB* and D-loop by the nucleotide p-distance (*Nei & Kumar, 2000*). Maximum likelihood (ML) phylogeny was constructed using the Hasegawa-Kishino-Yano (HKY) plus gamma (*CYTB*) and Tamura 3-parameter (T92) plus gamma model (D-loop) by the inbuilt model generator in MEGA5 (*Tamura et al., 2011*). An *Equus asinus* sequence (GenBank accession no.: NC001788) was used as an outgroup. Bootstrap analyses (1,000 replications) were used to assess the confidence of each node. According to the polymorphic sites, both haplotypes were assigned to the DomeTree (*Peng et al., 2015*) and D-loop haplotypes were also clustered into the haplogroups had been defined by *Jansen et al. (2002)*. A phylogenetic network based on merged *CYTB* and D-loop regions was constructed by use of a median-joining algorithm (*Bandelt, Forster & Röhl, 1999*) as implemented in the Network 4.1 program.

## RESULTS

Based on the sequence comparisons of the mitochondrial *CYTB* and D-loop markers, the Gidran horses showed high genetic variability. Twenty-three polymorphic sites were detected in the *CYTB* sequences, corresponding to two indels (e.g., insertion and deletion) and 21 single nucleotide polymorphisms (SNPs) with two transversions, and representing 3.35% of the analysed DNA sequence. Within the D-loop region, 26 polymorphic sites were found (one indel and 25 SNPs with a transversion) representing 12.9% of the analysed DNA sequence (Table 1). Both mtDNA regions were A/T rich with the following nucleotide frequencies: T, 27.7%; C, 31.5%; A, 27.3% and G, 13.5% in *CYTB* and T, 30.8%; C, 24.7%, A, 33.3% and G, 11.2% in D-loop. The A and T content was richer (55% and 64.1%) in both the *CYTB* and D-loop regions, respectively. These data were in accordance with the order of nucleotide composition in the vertebrate mitochondrial genome.

The calculated haplotype diversity (h) of the $CYTB$ and D-loop markers was $0.8735 \pm 0.011$ and $0.9136 \pm 0.008$, whereas the nucleotide diversity was $0.00472 \pm 0.00017$ and $0.02091 \pm 0.00068$, respectively. The paired genetic distances between the haplotypes were 0.001–0.013 ($CYTB$) and 0.005–0.063 (D-loop). All phylogenetic analyses were performed for both separate and combined mtDNA markers; a summary of the calculated genetic diversity parameters and the haplotypes of the Gidran mares are presented in Table 1.

Analysis of 686 bp of the $CYTB$ and 202 bp of the D-loop regions revealed 24 and 32 distinct haplotypes among the 31 Gidran mare families, the sequences of the $CYTB$ and D-loop haplotypes are available in GenBank database under accession no.: KT792934–KT792957 and KT818891– KT818922. The haplotypes differ from the reference sequences (GenBank accession nr.: X79547 and JN398377) by a maximum of 6 ($CYTB$) and 9 (D-loop) nucleotides. The polymorphic sites of both mtDNA markers are summarized in the Online (Table S1 and S2). $Ht1_{CYTB}$ ($n = 54$), $Ht2_{CYTB}$ ($n = 49$) and $Ht6_{CYTB}$ ($n = 44$) were the three most frequent haplotypes of the 24 $CYTB$ haplotypes, whereas seven ($Ht11_{CYTB}$, $Ht17_{CYTB}$, $Ht18_{CYTB}$, $Ht20_{CYTB}$, $Ht21_{CYTB}$, $Ht22_{CYTB}$, and $Ht24_{CYTB}$) were limited to only a single mare. The maximum likelihood tree representing the phylogenetic relationship among the 24 $CYTB$ haplotypes of the 250 Gidran mares is presented in Fig. 1. In the case of the D-loop haplotypes, $Ht6_{D\text{-loop}}$ ($n = 47$), $Ht16_{D\text{-loop}}$ ($n = 35$) and $Ht1_{D\text{-loop}}$ ($n = 25$) were the most common, whereas ten haplotypes ($Ht14_{D\text{-loop}}$, $Ht17_{D\text{-loop}}$, $Ht18_{D\text{-loop}}$, $Ht21_{D\text{-loop}}$, $Ht24_{D-loop}$, $Ht26_{D\text{-loop}}$, $Ht28_{D\text{-loop}}$, $Ht29_{D\text{-loop}}$, $Ht30_{D\text{-loop}}$, and $Ht31_{D\text{-loop}}$) were singletons. The BLAST search revealed six $CYTB$ ($Ht5_{CYTB}$, $Ht8_{CYTB}$, $Ht11_{CYTB}$, $Ht14_{CYTB}$, $Ht20_{CYTB}$, and $Ht21_{CYTB}$) and four new D-loop haplotypes ($Ht12_{D\text{-loop}}$, $Ht28_{D\text{-loop}}$, $Ht29_{D\text{-loop}}$, and $Ht32_{D\text{-loop}}$), which have not been published in NCBI database so far, and summarized in the Tables S1 and S2. The combined $CYTB$ and D-loop haplotypes could be clustered into ten (A, A1a, AB, B1, B1a, H, H-I, JK, M-N, and M-Q) haplogroups according to the DomeTree (Fig. S1) (Peng et al., 2015). Furthermore, D-loop haplotypes could also be assigned to seven of the major D-loop haplogroups defined by Jansen et al. (2002) with the following haplogroup distribution: A, 31%; B, 3%; C, 28%; D, 19%; E, 3%; F, 13% and G, 3% (Fig. 2). Among the 18 haplogroups (A-R) reported by Achilli et al. (2012), haplogroup E was not present in the Gidran native horses considered (Table S3 and S4).

Our additional key objectives were to identify errors in the Gidran stud book and to test the efficiency of the mtDNA markers for the separation of the mare families by haplotypes. Of the 31 mare families, 15 (48.4%) and 17 (54.8%) formed unique haplotypes according to the $CYTB$ and D-loop markers, respectively. Three mare families (borodi 14 and 18 and mezőhegyesi (1) could be separated exclusively by the $CYTB$, whereas four mare families (mezőhegyesi 2, 3, 19 and borodi (2) could be detected using only D-loop as a marker. Interestingly, individuals in the borodi 1 and 7 mare families formed a common haplotype when the two mtDNA markers were combined. On the other hand, seven mare families (mezőhegyesi 7, 8, and 21; borodi 5, 17, and 19; and népies 23) could be separated by the combination of $CYTB$ and D-loop markers. Five mare families (mezőhegyesi 5, 11, 13, and 14 and borodi 18) could not be isolated with either of the markers. A median-joining

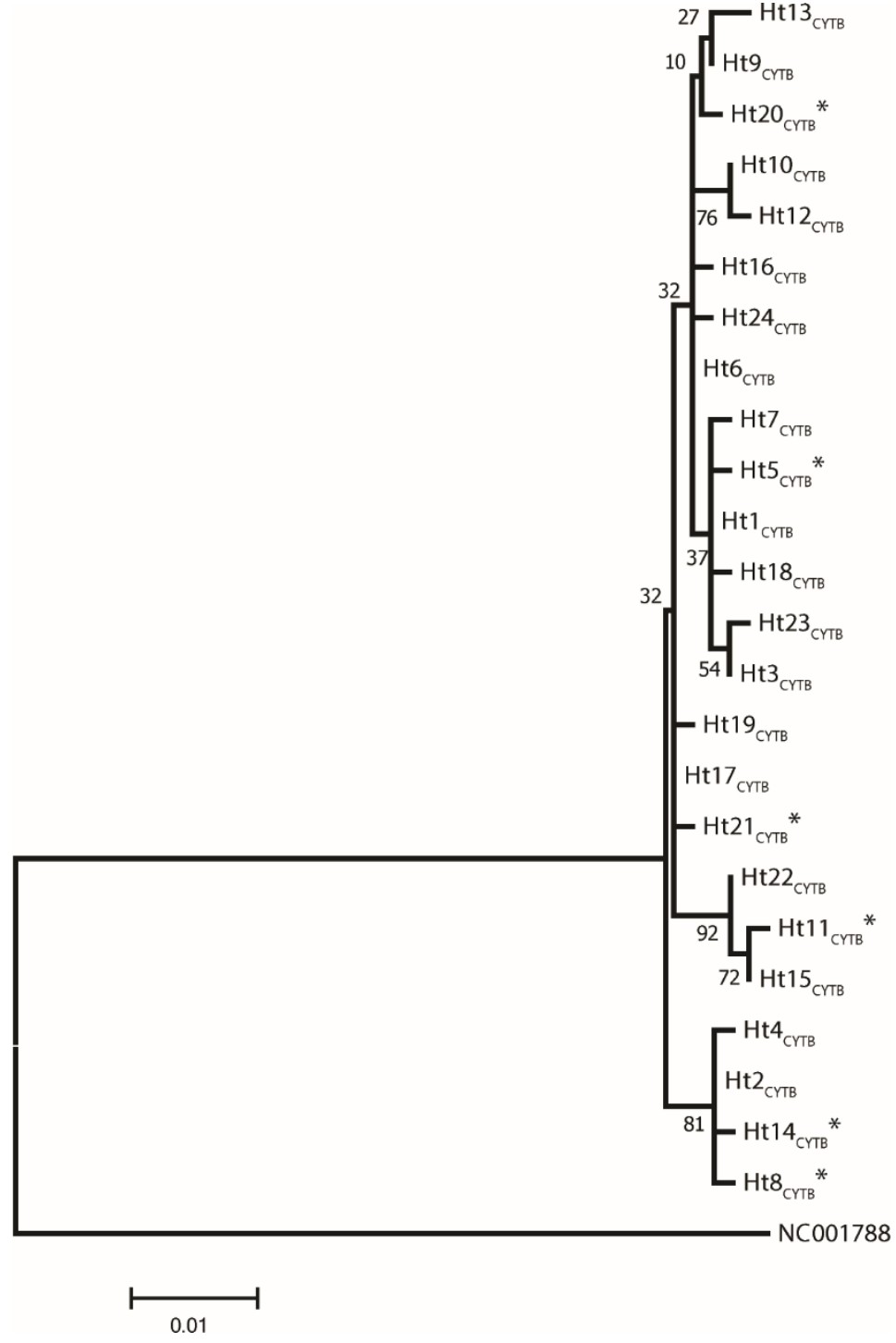

**Figure 1   A maximum likelihood tree represents the phylogenetic relationship among the 24 haplotypes based on 686 bp of protein coding mtDNA *CYTB* of 250 Gidran mares.** The phylogenetic tree was based on the Hasegawa-Kishino-Yano (HKY) model of evolution with gamma distribution of rates and 1,000 bootstrap replicates (*Hasegawa, Kishino & Yano, 1985*). The polymorphic sites considered relative to the X79547 reference sequence. Asterisks represent the new haplotypes.

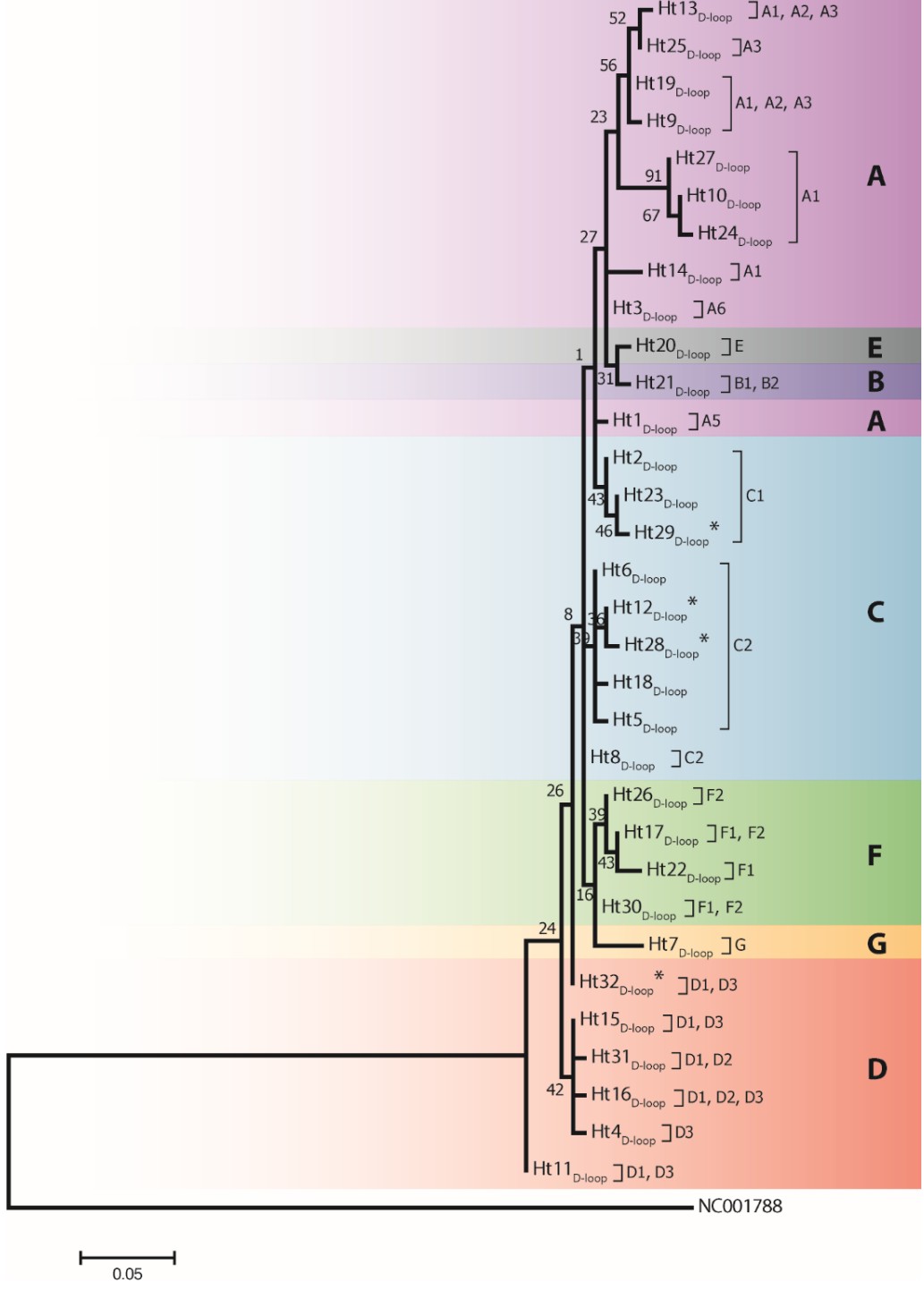

**Figure 2** **A maximum likelihood tree represents the phylogenetic relationship among 32 haplotypes based on 202 bp of mtDNA of the D-loop region of 246 Gidran mares.** The phylogenetic tree was based on the Tamura 3-parameter (T92) model of evolution with gamma distribution of rates and 1,000 bootstrap replicates (*Tamura et al., 2011*). Haplotypes were clustered into seven main haplogroups (A-F) described by *Jansen et al. (2002)*. The polymorphic sites considered relative to the X79547 reference sequence. Asterisks represent the new haplotypes.
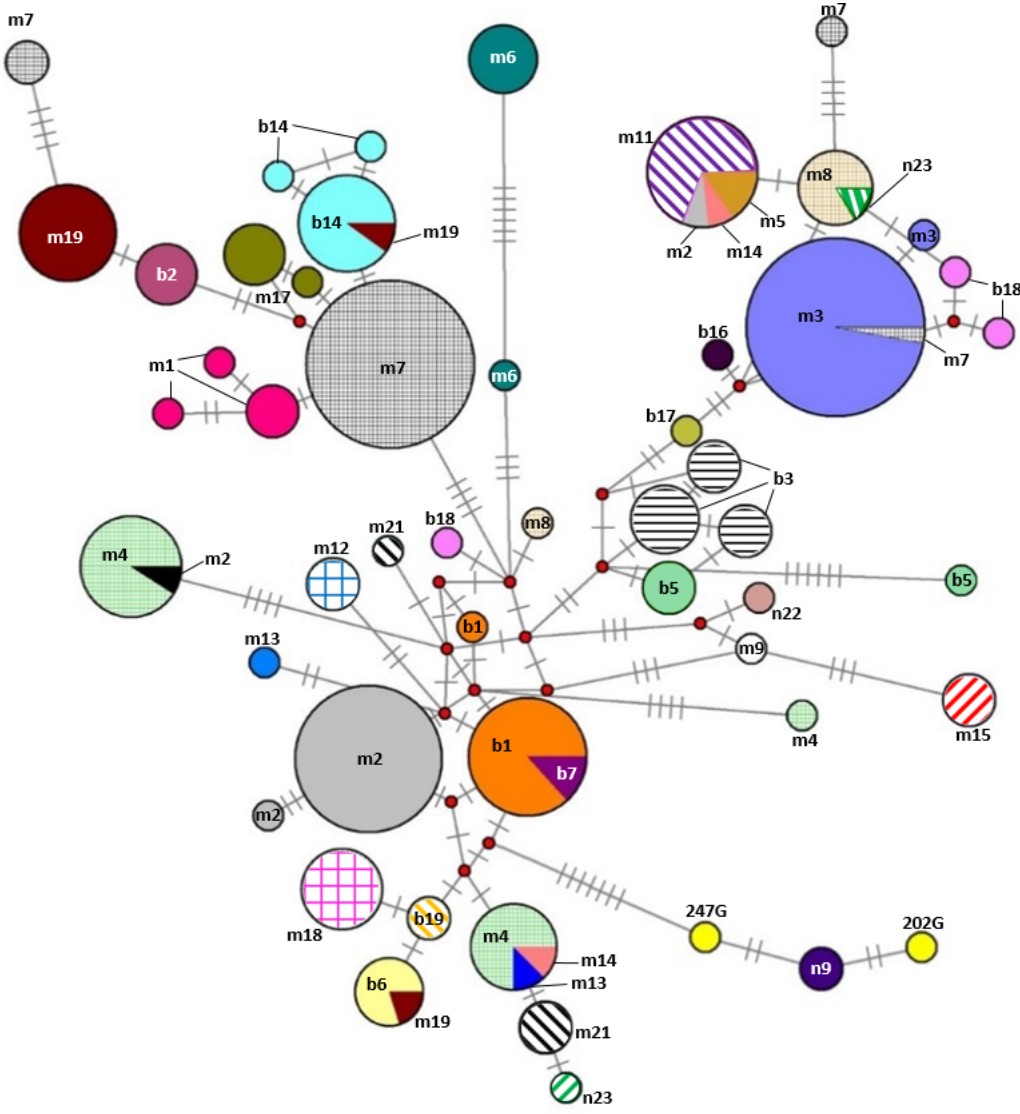

**Figure 3  Median-joining network based on combined nucleotide sequences (892 bp) of the mitochondrial *CYTB* and D-loop of 242 mares representing the 31 Gidran mare families.** Circle size is proportional to the sequence frequency and the colours represent different mare families. The number of transverse bars on the branches represent the number of nucleotide substitutions between the nodes. In the cases of 247G and 202G (yellow), the mare families are unknown. The abbreviations of the Gidran mare families are the following: m, mezőhegyesi; b, borodi; and n, népies.

network based on the combination of *CYTB* and D-loop regions was constructed by Network 4.1 and is shown in Fig. 3.

## DISCUSSION

The Gidran is a rare and endangered native Hungarian horse breed; therefore, the development of an effective conservation strategy is extremely urgent. A determination of phylogenetic relationships and the verification of stud book accuracy could be the first steps

in the maintenance of the genetic pool (*Bodó, Alderson & Langlois, 2005*). To date, contrary to the situation with the Hucul, which is another, but well-studied native Hungarian horse breed (*Czerneková, Kott & Majzlík, 2013*; *Georgescu et al., 2011*; *Kusza et al., 2013*), no data are available on the genetic structure and diversity of the Gidran. In this regard, through DNA sequence comparisons of *CYTB* and the D-loop, we investigated matrilineal diversity for Gidran horses in Hungary. Our research presents the first molecular phylogenetic study of the Gidran and covers approximately 87% (260/298) of the Hungarian Gidran mares from 31 traditional maternal lineages. Our mtDNA based data would tend to make the recent conservation strategies more successful to prevent genetic erosion in the Gidran.

First, we analysed the *CYTB* and D-loop mitochondrial DNA markers of the Gidran mares. According to our results, both markers reflected a strong genetic variety in Gidran. The presence of 24 and 32 mtDNA haplotypes with a total of 3.35% and 12.9% polymorphic sites in the *CYTB* and D-loop fragments reflect the broad genetic base of the Gidran maternal lines. Similar results have been found in the Hucul or Zemaitukai breeds which have relatively high nucleotide diversity despite suffering from severe bottlenecks during their histories (*Cothran, Juras & Macijauskiene, 2005*; *Kusza et al., 2013*).

The observed 32 D-loop haplotypes is similar in number to that reported for the Lusitano (27 haplotypes/145 horses) (*Lopes et al., 2005*), Lipizzan (37 haplotypes/212 horses) (*Kavar et al., 2002*) Arabian (27 haplotypes/200 horses) (*Bowling, Del Valle & Bowling, 2000*) and higher than in Kiso (7 haplotypes/ 136 horses) (*Takasu et al., 2014*). The calculated D-loop haplotype and nucleotide diversities are inconsistent with earlier horse mtDNA studies. The calculated nucleotide diversity was $0.02091 \pm 0.00068$, which is quite similar to the Iranian horse population ($0.02 \pm 0.000$) reported by *Moridi et al. (2013)*. This relatively high number indicates, that the Gidran is genetically more diverse than, for example, the Kerry Bog ($0.0155 \pm 0.0040$) and Sulphur Mustang breeds ($0.001 \pm 0.002$), but not more diverse than the Marwari ($0.03973 \pm 0.01262$) or Sorraia breeds ($0.104 \pm 0.012$) (*Devi & Ghosh, 2013*; *Luís et al., 2006*; *Prystupa et al., 2012*). Data from the *CYTB* sequences also confirmed the abundant genetic diversity of the Gidran. In the case of *CYTB*, the nucleotide diversity was lower in comparison to the D-loop, but it is similar to that observed in Chinese domestic horses, where the nucleotide diversity was between 0.00488 and 0.00186 while haplotype diversity was between 0.706 and 0.975 (*Yue et al., 2012*). *Qin et al. (2009)* sequenced a 1,140 bp length *CYTB* region in 22 Lichuan horses and also obtained high haplotype diversity and nucleotide diversity values (0.840 and 0.048, respectively). In a study, where *CYTB* was used for the characterization of maternal genetic origins and diversity of 323 horses from 13 Chinese indigenous breeds and 84 reference sequences from GenBank, 114 haplotypes were identified (*Yue et al., 2012*). The observed high D-loop and *CYTB* haplotype diversity confirm the multiple maternal origin of the Gidran, which might be explained by the fact that the recently known maternal lines were established by 16 founder mares (*Mihók, 2005*). These data suggest that although the Gidran is one of the smallest Hungarian horse population, the genetic diversity of the maternal lineage is preserved (*Takasu et al., 2014*).

BLAST showed that almost all the haplotypes found in the Gidran samples are identical to other domestic horse haplotypes in GenBank except for six *CYTB* and five D-loop

haplotypes that have not been described for any other horse breed yet. Because uncommon haplotypes have an increased risk of extinction (*Lopes et al., 2005*), these distinct haplotypes support the importance of maintaining rare individuals and also emphasize the genetic diversity of the Gidran.

Pedigree analysis also plays a key role in breeding programmes, which aim to maintain genetic diversity of endangered populations (*Bokor et al., 2013*). Our further aim was to recognize overlapping haplotypes among mare families and thereby detect incidental errors in the Gidran stud book. Using the mtDNA markers alone was not sufficiently effective because neither *CYTB* nor D-loop sequences made the alignment of the haplotypes to each of the mare families reasonably acceptable. To improve the efficiency of our data, starting from the study of *Pedrosa et al. (2005)*, a phylogenetic analysis with the combined *CYTB* and D-loop markers was also performed (Fig. 3). Although, the results showed several inconsistencies in the distribution of the 49 common *CYTB* and D-loop haplotypes within the 31 mare families, we found this approach more effective to screen for registry errors rather than using only one mtDNA marker.

Altogether, pedigree records were problematic in seven mares (2.89%) registered in the Gidran stud book. This number is small compared with Lipizzan and Polish horses where the discrepancies were 11% between pedigree data and mtDNA haplotypes in studies on both breeds (*Głażewska et al., 2007*; *Kavar et al., 2002*). The data collected in the present study indicates the management of the Gidran stud book was appropriate over the years. The seven problematic mares belonged to a different cluster (independent of the DNA markers and their combination) than suggested according to the Gidran stud book. Several reasons could explain the described inconsistencies. However, errors may have been made in the management of the stud book because the approximately 200 years of existence of the maternal line is a very short time for the formation of a distinct haplotype of each mare family (*Devi & Ghosh, 2013*).

Four horses did not possess their own cluster. Among them, two animals from the borodi 7 mare family formed a common cluster with borodi 1. Furthermore, two individuals of the mezőhegyesi 5 mare family shared a haplotype with animals of mezőhegyesi 11. These discrepancies are in concordance with the mare family's history (*Mihók, 2005*). Individuals of borodi 1 and 7 shared the same haplotype. Possibly, mare families sharing the same haplotypes belong to the same mare family, which is, as a consequence of the incomplete pedigree data, now split into different mare families (*Kavar et al., 2002*). In contrast, based on the analysis of the two mitochondrial markers separately and combined, individuals of the mezőhegyesi 4 mare family formed two distinct haplotypes. This observation was confirmed by the available historical data, which suggests that the mezőhegyesi 4 mare family diverged over the years (*Mihók, 2005*).

## CONCLUSIONS

In addition to providing our first insight into the maternal mitochondrial diversity in the rare native Hungarian Gidran breed, this study also provided the opportunity to compare molecular genetic results with stud book data. Our key finding was that high matrilineal

diversity was observed in the Gidran breed using both *CYTB* and D-loop markers despite the history of the breed. Moreover, the obtained haplotypes are mostly consistent with stud book's mare families, particularly when the two mtDNA markers were combined. Gidran breeders are recommended to take this information into account in the future.

### Funding

The authors received no funding for this work.

### Competing Interests

The authors declare there are no competing interests.

### Author Contributions

- Nikolett Sziszkosz conceived and designed the experiments, performed the experiments, analyzed the data, wrote the paper, prepared figures and/or tables.
- Sándor Mihók and András Jávor conceived and designed the experiments, contributed reagents/materials/analysis tools, reviewed drafts of the paper.
- Szilvia Kusza conceived and designed the experiments, analyzed the data, contributed reagents/materials/analysis tools, reviewed drafts of the paper.

### Animal Ethics

The following information was supplied relating to ethical approvals (i.e., approving body and any reference numbers):

All horses in this study were client-owned on which no harmful invasive procedure was performed; and there was no animal experimentation according to the legal definitions in Europe (Subject 5f of Article 1, Chapter I of the Directive 2010/63/UE of the European Parliament and of the Council).

### DNA Deposition

The following information was supplied regarding the deposition of DNA sequences:

GenBank accession numbers of CYTB haplotypes: KT792934, KT792935, KT792936, KT792937, KT792938, KT792939, KT792940, KT792941, KT792942, KT792943, KT792944, KT792945, KT792946, KT792947, KT792948, KT792949, KT792950, KT792951, KT792952, KT792953, KT792954, KT792955, KT792956, KT792957

CYTB:

http://www.ncbi.nlm.nih.gov/popset/951669370

GenBank accession numbers of D-loop haplotypes:

KT818898, KT818892, KT818893, KT818894, KT818895, KT818896, KT818897, KT818891, KT818899, KT818900, KT818901, KT818902, KT818903, KT818904, KT818905, KT818906, KT818907, KT818908, KT818909, KT818910, KT818911, KT818912, KT818913, KT818914, KT818915, KT818916, KT818917, KT818918, KT818919, KT818920, KT818921, KT818922

D-loop:

http://www.ncbi.nlm.nih.gov/popset/930420559.

## Data Availability

The research in this article did not generate any raw data.

## Supplemental Information

Supplemental information for this article can be found online at http://dx.doi.org/10.7717/peerj.1894#supplemental-information.

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
