# Peer review of "Genetic diversity of the Hungarian Gidran horse in two mitochondrial DNA markers"

_PeerJ, doi:10.7717/peerj.1894_

## Round 0.1 · original submission · Major Revisions

Please respond to all the reviewer comments. The reviewers have asked for a number of clarifications, which I am confident you will be able to make within the timeframe.

Reviewer 1 ·

Basic reporting

This is an interesting study that I think should be published. However, there are a few points of concern that have to be addressed before the article should be considered for publication:

1) Line 70: "which encode 13 proteins that are part of 5 protein complexes" is wrong and should be changed to "which encode 13 polypeptides that form part of 4 protein complexes (CI, CIII, CIV and CV) of the OXPHOS system".

2) Ethics might be a concern. Although the authors state on Line 90 that "No approval required for the type of research undertaken", elaborating on this point is recommended to clarify if the ethics approval is not required by their Institution / country etc. as ethics approval might be essential for this type of study in other countries.

Experimental design

3) Major concern: The authors give full detail how the DNA was isolated and amplified, but almost no information on the sequencing with the only information on line 112: "..and then sequenced directly with the same primers used form amplification". How was the DNA sequenced, where, by whom? Also, no information are given on quality control samples that were included to verify that the PCR correctly amplified the DNA (incorporating the correct nucleotides in the amplicons) or that the sequencing of a control sample was correct, resulting in the correct sequence. Without this information convincing the reader that the data is reliable, the rest of the paper becomes suspect. I would suggest adding the information to the paper

Validity of the findings

4) 250 Samples were collected, while data is reported for 250 (CYTB), only 246 for the D-loop and lastly only 242 for CYTB and D-loop combined. Why the difference in numbers. Where there issues with the PCR and sequencing of all samples? Why did the authors not explain this discrepancy? This links with the concern about the quality control of the study.

Additional comments

In general the language is acceptable, although it is recommended that a native English speaker have a look at the language before resubmitting the paper

Reviewer 2 ·

Basic reporting

1) I suggest the authors adding one figure to exhibit the relationship of 31 families in the studbook. It will be helpful for readers.

2) Line 120: Gene Bank should be GenBank.

3) Line 248 - 249: D-loop and CYTB gene are mtDNA markers. The sentence should be revised.

4) The references in Table S3 need corrections. Some references have been published.

Experimental design

No Comments.

Validity of the findings

1) The old reference sequence X79547 should be abandoned due to containing errors (Achilli et al. 2012. PNAS. 109:2449-54). Instead, the authors should adopt JN398377 as reference. And Table S1 should be revised.

2) The standard mtDNA haplogroup tree together with lineage nomenclature for horse is available in DomeTree (http://www.dometree.org). I suggest the authors refer it to avoid potential confusions and chaos in mtDNA haplogoruping.

3) The authors constructed the ML tree. Nevertheless, the bootstrap values were relative low, most likely because of short length of sequenced fragments. Referring to the available horse mtDNA haplogroup tree, the authors can assign mtDNA lineages into specific haplogorup in terms of scored variants of D-loop and CYTB. This strategy has been proven to be effective and powerful (Peng, et al. 2015. PNAS. 112: E1970-71).

5) The authors should declare unpublished data used in this manuscript, such as how to get it.

6) Line 210 - 213: Numbers of haplotypes are depending on sample sizes. How about the sample sizes in those reported studies? The simplified comparison is improper.

---

## Round 0.2 · Minor Revisions

Let me apologize for the delay, one of the reviewers was delayed in submitting their review, we only received it yesterday. The most important points made by the reviewers are addressed. But please consider the clarfication made by reviewer 2 regarding the supplemental data. Once this has been considered I will be happy to approve the manuscript for publication.

Reviewer 1 ·

Basic reporting

Commented on this during the first review

Experimental design

Commented on this during the first review

Validity of the findings

Commented on this during the first review

Additional comments

The authors addressed all the issued I raised during the first review successfully. Thus I recommend that the paper be published in it's current form

Reviewer 2 ·

Basic reporting

No Comments.

Experimental design

No Comments.

Validity of the findings

No Comments.

Additional comments

1) Are the individual codes the same in Table S1 and Table S2?

2) The authors misunderstood the previous 7) comments. I suggest to combining D-loop and CYTB sequences to define the haplotypes and then to classify the haplogroups. For instance, for the individuals both sequenced D-loop and CYTB regions, the authors can consider the scored variants both from 202 bp and 686 bp. It is because mtDNA is haploid marker which is in completely linkage disequilibrium. In terms of the combined information, Figure S1 and Figure S2 can be integrated into one figure. In this figure, the reference JN398377 should be present.

---

## Round 0.3 · accepted · Accept

You have addressed all the comments raised by the reviewer. I will recommend that the manuscript be accepted.